# Growth Differentiation Factor 15 as a Predictor of the No-Reflow Phenomenon in Patients with ST-Segment Elevation Myocardial Infarction

**DOI:** 10.3390/jcm12010245

**Published:** 2022-12-29

**Authors:** Marcin Kożuch, Maciej Południewski, Emil Julian Dąbrowski, Ewa Tarasiuk, Sławomir Dobrzycki

**Affiliations:** 1Department of Invasive Cardiology, Medical University of Bialystok, 24A Sklodowskiej-Curie St., 15-276 Bialystok, Poland; 2Department of Cardiology, Medical University of Bialystok, 24A Sklodowskiej-Curie St., 15-276 Bialystok, Poland

**Keywords:** growth differentiation factor 15, myocardial infarction, no-reflow phenomenon

## Abstract

Growth differentiation factor 15 (GDF-15) and the no-reflow phenomenon are predictors of mortality after ST-segment elevation myocardial infarction (STEMI). We aimed to assess the relation between GDF-15 concentration on admission and the no-reflow phenomenon. The study was conducted prospectively among 80 consecutive STEMI patients who underwent primary PCI. No-reflow was defined as a corrected TIMI frame count > 27 and myocardial blush grade < 3 after PCI. GDF-15 was measured on admission. We assessed long-term (1.3 years) total mortality and the risk factors of no-reflow. The mean age was 65 (SD 12) years. Mortality rates were 2.5% and 7.5% for in-hospital and long-term observations, respectively. No-reflow occurred in 24% of patients. A negative correlation between TIMI flow after PCI and GDF-15 concentration (R = −0.2540, *p* = 0.023) was found. Receiver operating characteristic (ROC) analysis revealed GDF-15 as a predictor of no-reflow (AUC-0.698, 95%CI-0.552–0.843, *p* < 0.05). The multivariate logistic regression analysis revealed that the risk factors for no-reflow occurrence were higher age, a concentration of GDF-15 > 1503 pg/mL, lower systolic blood pressure, and higher troponin I concentration on admission. A higher concentration of GDF-15 can be used as an additional marker of ischemia/reoxygenation injury, subsequent no-reflow phenomenon, and worse long-term outcomes in patients with STEMI.

## 1. Introduction

Growth differentiation factor 15 (GDF-15) is a member of the transforming growth factor superfamily [1]. The GDF-15 is expressed in all organs and tissues which suggests its important functions. However, the precise role of GDF-15 is still not fully recognized. GDF-15 is involved in apoptotic and inflammatory pathway regulation and its expression is upregulated in many different pathological conditions, such as cardiovascular, pulmonary, and renal diseases. Interestingly, GDF-15 can perform different functions under various conditions [2]. Despite the incomplete understanding of GDF-15s role, we are already aware of its high predictive value for worse prognosis in patients presenting higher concentrations of GDF-15 during myocardial infarction [3,4,5]. The relationship between higher GDF-15 concentration and poor outcomes in this group of patients is unclear, especially taking into consideration the potentially protective functions of GDF-15 described in experimental models of myocardial infarction [6,7]. Thus, we decided to establish if there is any correlation between GDF-15 concentration and disturbances affecting microcirculatory reflow in patients with myocardial infarction after primary percutaneous coronary intervention (PCI).

## 2. Materials and Methods

### 2.1. Patients

The study was conducted prospectively among 80 consecutive STEMI patients who underwent primary PCI in our center between 2012 and 2014. GDF-15 concentration was measured on admission to the hospital in the peripherally collected blood samples. Standard commercial ELISA kits were used (Human GDF-15 Immunoassay Quantikine, R&D) for the measurements. Angiographically established no-reflow was defined as a corrected TIMI frame count exceeding 27 and myocardial blush grade less than 3 immediately after the procedure of PCI [8,9]. All angiograms were analyzed independently by two experienced investigators and blinded to all clinical data investigators. Clinical, angiographic, procedural, and biochemical characteristics were taken into consideration during defining predictors of no-reflow. Pharmacotherapy was administered according to the European Society of Cardiology guidelines binding at that time (Appendix A). All patients received a loading dose of clopidogrel, ticagrelor, or prasugrel, and aspirin. Intravenous unfractionated heparin tailored to the weight of the patient was administered before or during the procedure. Total mortality was estimated during hospitalization and at follow-up. Informed consent was obtained from each patient. The study protocol conforms to the ethical guidelines of the 1975 Declaration of Helsinki and was approved by the Bioethics Committee of the Medical University of Białystok.

### 2.2. Follow-Up

The mean follow-up was 461 (SD 248) days. Follow-up data were collected from the Voivodship Office. Information concerning total mortality was obtained in 100% of cases. The Office provides information concerning the death of a person, however, the cause of death remains unknown. 

### 2.3. Statistical Analysis

Distributions of all variables were assessed with the Kolmogorov–Smirnov test. The correlations between constant variables were calculated with Spearman or Pearson tests depending on the statistical distribution. Correlations between dichotomous variables were analyzed with the Chi2 test. Multivariate analysis was performed with a logistic regression test. Receiver operating characteristic (ROC) curves analysis was used to establish the predictive value of GDF-15 for the no-reflow phenomenon occurrence. The results are presented as mean values with standard deviation, median with interquartile range (IQR) according to variation distribution, or as percentages presenting relative frequency. We considered a *p* value < 0.05 as statistically significant. Statistical analysis was performed with Statistica 10.0 program (StatSoft, Inc. Tulsa, AK, USA).

## 3. Results

The mean age of the studied population was 64.75 (11.72) years, and 73% of the study population were males. The study population consisted of 5% of patients with previous myocardial infarction and 6% of patients with a history of revascularization (Appendix A). No-reflow was found in 24% of patients (N = 19). Subjects with no-reflow were older, had a longer duration of symptoms, lower e-GFR and blood pressure, and higher levels of myocardial necrosis markers on admission (Appendix A, Table 1). Moreover, we found more frequent pre-dilatation use in the no-reflow group of patients (Table 2). Detailed history, clinical, biochemical, and angiographic characteristics, and comparison of no-reflow and reflow subjects are presented in Table 1, Table 2 and Table 3 and Appendix A.

Mortality rates were 2.5% (N = 2) and 7.5% (N = 6) for in-hospital and long-term observation, respectively. No-reflow was associated with a worse hospital (13% vs. 2%, *p* = 0.05) and long-term mortality (21% vs. 3%, respectively, *p* < 0.01). GDF-15 concentration and no-reflow phenomenon were strongly correlated with the risk of long-term mortality (consecutively R = 0.4476, *p* < 0.001 and R = 0.2872, *p* = 0.01).

Patients with no-reflow had a higher concentration of GDF-15 on admission than reflow ones (1246.08 (1145.43) vs. 1075 (711.6), respectively, *p* < 0.03; Table 2). A negative correlation between TIMI flow after PCI and GDF-15 concentration (R = −0.2540, *p* = 0.023) was found. The analysis of ROC curves revealed GDF-15 as a predictor of no-reflow; the area under the curve was significantly higher than for the random model (AUC 0.698, 95%CI 0.552–0.843, *p* < 0.05, Figure 1). The best predictive value of no-reflow for GDF-15 concentration was defined as 1503 pg/mL (sensitivity 63%, specificity 84%, Figure 2). The multivariate logistic regression analysis revealed that the risk factors for no-reflow occurrence were higher age, the concentration of GDF-15 > 1503 pg/mL, lower systolic blood pressure, and higher troponin I concentration on admission (Table 4).

## 4. Discussion

No-reflow, as a clinical manifestation of ischemia/reoxygenation injury, is found in many patients with reperfused STEMI. No-reflow phenomenon affecting patients with myocardial infarction treated invasively diminishes the potential benefits of the treatment [10]. In this study, we have found a correlation between GDF-15 and disturbances in microcirculatory reflow in patients with STEMI. No-reflow as a disturbance of reflow in microcirculation persists for a long time after reperfusion, enlarges the size of the infarcted area, and affects the outcome [11,12,13]. It promotes negative remodeling of the myocardium and thus may directly trigger the development or lead to the progression of congestive heart failure [14,15,16]. All mechanisms leading to no-reflow are still not fully explained. Some studies linked the prolonged ischemia time with microcirculatory damage and the no-reflow phenomenon [17]. It is consistent with our study as the time pain to door was significantly longer in the no-reflow group (Appendix A). The correlation between higher GDF-15 concentration and no-reflow found in this study may constitute a bridge connecting poor outcomes of patients with STEMI and elevated GDF-15 accumulation. Various white cells seem to play a crucial role in reperfusion injury leading to the no-reflow phenomenon [18]. Thus, GDF-15 as a cytokine of an important function in white cells signal transmission can be found in higher concentrations in infarcted myocardium. Consequently, higher preformation and release of GDF-15 during an ischemic period of infarction can be directly related to inflammatory-cell accumulation participating in the microcirculation injury. Thus, the observed higher concentration of GDF-15 may be increased secondarily to larger infarction and activation of inflammatory cells involved in microcirculation injury during ischemia followed by reperfusion. Therefore, increased levels of GDF-15 should be interpreted as a marker of higher stress in infracted myocardium rather than the factor promoting myocardial injury and the no-reflow phenomenon. This theory is supported by the results of studies published by Kempf T. et al. and Zhang M. et al. [6,19]. The authors support the conception of a protective role of GDF-15 in the heart during ischemia/reperfusion injury. In another study, Kempf T. et al. found that GDF-15 is essential in the prevention of myocardial rupture after myocardial infarction in mice [7]. Seemingly in opposition to this observation, Dominguez-Rodriguez A. found that higher levels of GDF-15 were associated with negative remodeling of the left ventricle after STEMI [20]. Elevated levels of GDF-15 in this study should be interpreted as a result of higher stress in reperfused myocardium and more extensive injury rather than a straight factor leading to a worse outcome. Despite the precise role of GDF-15 in these processes, the observed correlation between its concentration and angiographic signs of no-reflow is uncontested. Regardless of this relationship, the detailed role of GDF-15 in the mechanism of no-reflow and administration of possible pharmacological intervention requires more extensive studies in this field of knowledge. 

The recent individual patient meta-analysis that investigated GDF-15 prognostic performance in atherosclerotic cardiovascular disease revealed that its baseline concentration is associated with the incidence of major adverse cardiovascular events (MACE) and cardiovascular death [21]. However, it had no prognostic association with future myocardial infarction (MI) and stroke in patients with acute coronary syndrome (ACS). The prognostic value of GDF-15 was previously reported in patients with non-ST-elevation acute coronary syndromes (NSTE-ACS) [22]. Khan S. et al. reported that GDF-15 was a significant predictor of death in non-ST-elevation myocardial infarction (NSTEMI) but not in STEMI patients [23]. This may be explained by the different methods of sample collection—the post-reperfusion blood was analyzed. We have found a correlation of GDF-15 concentration with higher long-term mortality in patients with STEMI, which is consistent with previously published studies with methodologies similar to ours [24,25]. The no-reflow occurrence was also a risk factor for poor long-term outcomes in our analysis. In addition, GDF-15 levels correlated negatively with an ejection fraction of the left ventricle (R = −0.2207, *p* = 0.049), which is a well-recognized risk factor for poor prognosis. These results support the conception of GDF-15 as a predictor of no-reflow and larger myocardial injury leading to poor outcomes. However, this observation should be interpreted cautiously as we found GDF-15 concentration as an independent from ejection fraction to be a significant risk factor for long-term mortality in multiple regression analysis (β 0.2722, SE of β 0.1033, *p* = 0.03).

Our research brings GDF-15 into sharp focus as a potential marker of ischemia/reoxygenation injury and subsequent no-reflow phenomenon. Due to the fact that the assessment of GDF-15 and other biomarkers’ concentrations requires highly-specialized laboratories and time-consuming methods, their clinical usefulness is currently low. However, there are already rapid bedside tests for the quantitative determination of protein levels (e.g., C-reactive protein). Further research may contribute to the development of such tests for GDF-15, as opposed to a time-absorbing standard laboratory assay. In the future, this would improve the decision-making process and influence rapid initiation of the treatment that protects against the no-reflow phenomenon, such as the administration of GP IIb/IIIa inhibitors.

The basal limitation of the presented analysis is the relatively small number of patients. However, this was a prospective study and the number of patients included in the analysis was enough to find statistically significant correlations which additionally supports the strengths of observed findings. We made only one measurement of GDF-15 on admission. Further measurements in a number of time intervals would enrich our knowledge of dynamic changes of GDF-15 in the studied population. Such assessments would provide additional data about the kinetics of GDF-15 after reperfusion. Nevertheless, the main aim of our study—observation of the correlation between GDF-15 levels on admission and no-reflow occurrence—was possible with only one measurement of GDF-15 concentration. The method used for the no-reflow assessment can be interpreted as another limitation of the presented study. There are many different methods defining the no-reflow phenomenon [8,9,13]. Mostly accurate but less accessible methods are MRI and contrast echocardiography [26,27]. However, ECG and angiographic analyses seem to be the simplest and generally available ones. We used angiographic analysis, which is an acceptable and widely used method providing the possibility to evaluate microcirculatory flow during angiography. Moreover, we observed in this study a correlation between GDF-15 concentration and the lack of 50% ST-segment resolution in ECG post PCI (R = 0.2221, *p* = 0.05). However, to improve the clarity of the methodology and the meaning of the finding, we decided to reduce our analysis to an angiographic assessment of no-reflow. Lastly, our study does not report case-specific mortality, which prevented us from assessing the prognostic value of GDF-15 on CV mortality.

## 5. Conclusions

Concluding our prospective study results, a higher concentration of GDF-15 on admission can be used as an additional marker of ischemia/reoxygenation injury and subsequent no-reflow phenomenon in patients with STEMI treated with primary PCI.

## Figures and Tables

**Figure 1 jcm-12-00245-f001:**
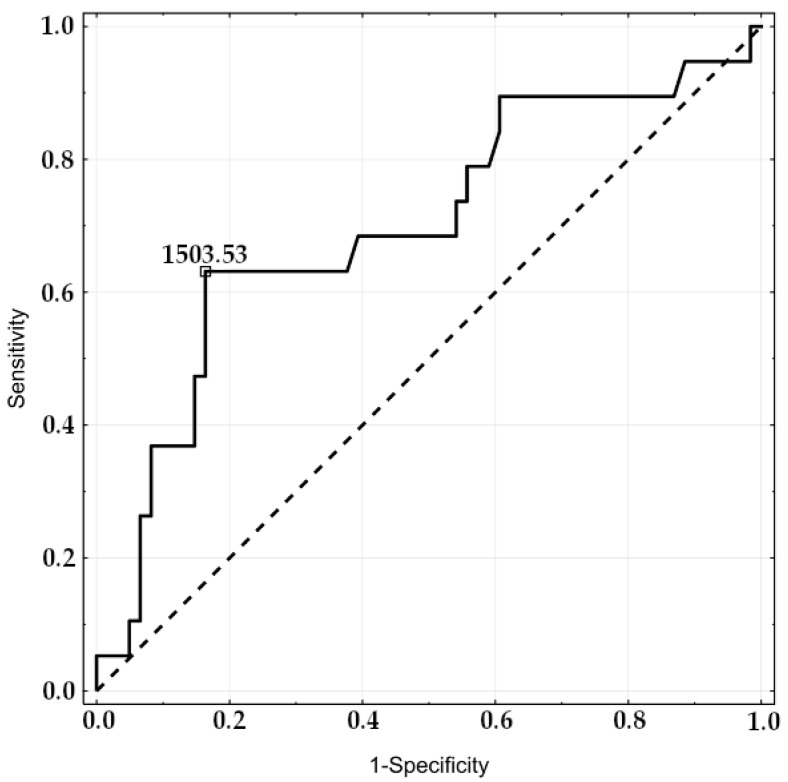
The receiver-operating characteristic curve of GDF-15 concentration as a predictor of no-reflow (AUC 0.698, 95% CI 0.552–0.843, *p* < 0.05).

**Figure 2 jcm-12-00245-f002:**
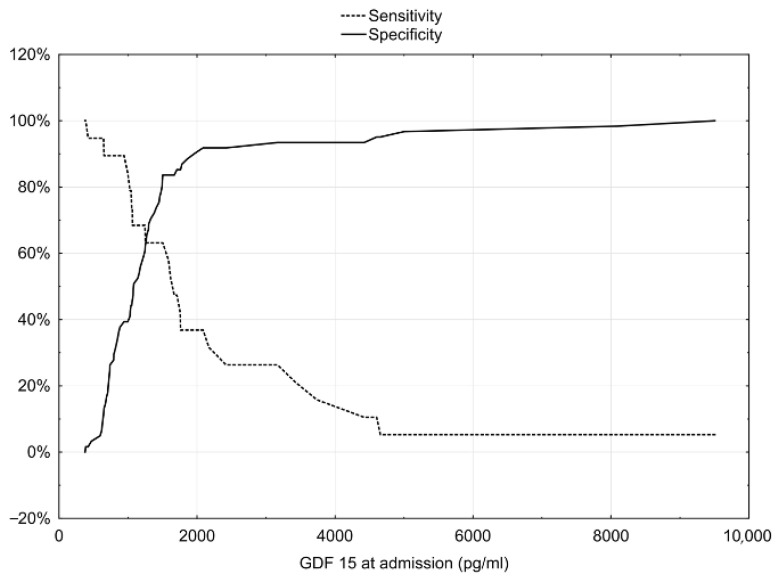
Sensitivity and specificity for GDF-15 concentration as a predictor of no-reflow.

**Table 1 jcm-12-00245-t001:** Biochemical characteristics.

	All Patients N = 80	No-Reflow Group N = 19	Reflow Group N = 61	*p*
TnI at admission (mg/dl)	0.55 (6.27)	1.0 (13.34)	0.22 (2.26)	<0.001
CK at admission (U/L)	178 (411)	168 (927)	188 (209)	<0.001
CK-MB at admission (IU/L)	32.5 (40.5)	33 (173)	30 (25)	<0.001
Maximal TnI (mg/dl)	50 (26.5)	50 (2)	42.9 (29.4)	0.167
Maximal CK (IU/L)	2475.39 (2037.08)	3125.21 (2343.67)	2272.98 (1908.02)	0.112
Maximal CK-MB (IU/L)	324 (404)	418 (318)	447.93 (999.96)	0.716
Glycemia on admission (mg/dl)	135 (72)	128 (76)	138.5 (68.5)	0.597
Total cholesterol	210.83 (45.54)	204.84 (55.78)	212.76 (42.10)	0.513
LDL cholesterol	145.41 (44.66)	140.84 (45.06)	146.88 (44.82)	0.611
HDL cholesterol	49.45 (13.69)	47.26 (17.49)	50.15 (12.32)	0.427
TG cholesterol	206 (68)	203 (77)	211 (65)	0.889
GDF-15 at admission (pg/mL)	1170.05 (808.43)	1246.08 (1145.43)	1075 (711.6)	0.023

TnI—troponin I, CK—creatine kinase, CK-MB—creatine kinase myocardial band, LDL—low-density lipoprotein, HDL—high-density lipoprotein, TG—triglycerides, GDF-15—growth differentiation factor 15.

**Table 2 jcm-12-00245-t002:** Angiographic and procedural characteristics.

	All Patients N = 80	No-Reflow Group N = 19	Reflow Group N = 61	*p*
Radial access	94%	84%	97%	0.05
Left descending artery as IRA	52%	62%	49%	0.365
Right coronary artery as IRA	36%	28%	39%	0.537
Circumflex artery as IRA	12%	10%	12%	0.544
% of stenosis	100 (1)%	100 (0)%	100 (1)%	0.052
TIMI flow 0 before PCI	70%	90%	64%	0.034
Thrombectomy	57.50%	74%	52%	0.105
Predilatation	61.25%	84%	54%	0.018
No. of stents implanted	1.25 (0.61)	1.16 (0.60)	1.28 (0.61)	0.451
DES implantation	81.25%	79%	82%	0.822
BMS implantation	22.50%	16%	25%	0.429
Stent diameter (mm)	3.44 (0.5)	3.25 (0.5)	3.5 (0.5)	0.451
Stent length (mm)	20 (12.5)	20 (11)	20 (183.5)	0.722
SYNTAX Score at admission	17.78 (8.55)	17.50 (6.91)	17.86 (9.05)	0.874
No-reflow	24%	-	-	-

IRA—infarction-related artery, PCI—percutaneous coronary intervention, DES—drug-eluting stent, BMS—bare metal stent, TIMI—thrombolysis in myocardial infarction.

**Table 3 jcm-12-00245-t003:** Past medical history data.

	All Patients N = 80	No-Reflow Group N = 19	Reflow Group N = 61	*p*
Coronary artery disease	16%	26%	13%	0.177
Myocardial infarction	5%	5%	5%	0.952
PCI/CABG	6%	5%	7%	0.841
Smoking	46%	21%	54%	0.011
Arterial hypertension	64%	53%	67%	0.253
Diabetes mellitus	18%	26%	15%	0.252
Hyperlipidemia	78%	63%	82%	0.088
Chronic kidney disease	16%	17%	16%	0.982

CABG—coronary artery bypass grafting, PCI—percutaneous coronary intervention

**Table 4 jcm-12-00245-t004:** Risk factors for no-reflow phenomenon in multivariate regression analysis.

	β	SE of β	B	SE of B	t (70)	*p*
Age (years)	0.3849	0.1458	0.0141	0.0053	2.6404	0.0102
Time pain to door (minutes)	0.0639	0.1224	0.0001	0.0001	0.5220	0.6033
TIMI flow before PCI	−0.1390	0.1012	−0.0547	0.0398	−1.3734	0.1740
GDF-15 level > 1503 pg/ml	0.2381	0.1046	0.2303	0.1011	2.2769	0.0259
e-GFR (ml/minute)	0.2274	0.1457	0.0030	0.0019	1.5606	0.1231
RRs on admission (mmHg)	−0.2904	0.1277	−0.0049	0.0022	−2.2741	0.0260
RRd on admission (mmHg)	0.0567	0.1264	0.0016	0.0036	0.4486	0.6551
TnI on admission (mg/dl)	0.2750	0.1213	0.0102	0.0045	2.2674	0.0265
Predilatation	0.0758	0.0982	0.0662	0.0858	0.7717	0.4429

TIMI—thrombolysis in myocardial infarction, PCI—percutaneous coronary intervention, GDF-15—growth differentiation factor 15, e-GFR—estimated glomerular filtration rate, RRs—systolic blood pressure, RRd—diastolic blood pressure, TnI—troponin I.

## Data Availability

Data available from corresponding author upon reasonable request.

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
