# Peer review of "Growth Differentiation Factor 15 as a Predictor of the No-Reflow Phenomenon in Patients with ST-Segment Elevation Myocardial Infarction"

_jcm, 2022, doi:10.3390/jcm12010245_

Round 1

Reviewer 1 Report

Comments to the Authors

The authors conducted a prospective, single-center, observational study (n=80), in which the clinical usefulness of GDF-15 was tested. They showed that the higher GDF-15 level was associated with no-reflow phenomenon during the primary PCI procedures. GDF-15 may be a promising biomarker in the field of cardiovascular disease and thus, the present paper may be of clinical interest. However, there are lots of points that should appropriately addressed by the authors.

1.       The authors state that their study is prospectively performed. Thus, the trial registration number should be shown. If not registered, the present study should be considered as if it was a retrospective study. In addition, this reviewer does not understand why the study period was between 2012 and 2014 (i.e. outdated). GDF-15 may have been evaluated using blood samples for some reasons? It is also peculiar that the mean follow-up duration was “only” 461 days against the fact that the study period ended in 2014 (8 years ago).

2.       No-reflow phenomenon in the present study was defined as a corrected TIMI Frame Count excessing 27 and Myocardial Blush Grade less than 3 “after” PCI, and the incidence was reported to be 24%. This reviewer wonders if the primary event included no-reflow phenomenon “during” the primary PCI procedures, or that “after” PCI (i.e. final TIMI flow). If authors included only no-reflow phenomenon “after” PCI, the rate (24%) sounds extremely high. The definition of no-reflow phenomenon, including the timing for evaluation, should be clear.

3.       The elevated TnI, CK, and CK-MB levels at admission in the no-reflow group may be attributable to the longer onset-to-arrival and onset-to-balloon times. According to the CK level at admission, most of patients in the no-reflow group probably came to the hospital >12 hours after symptom onset. Onset time seems crucial data. Table S2 should be included in the main manuscript. By the way, this reviewer wonders if TnI, CK, and CK-MB levels are expressed as median (IQR), because of the skewed distribution.

4.       NS for p values should be avoided.

5.       A recent, large-scale, patient-level meta-analysis investigating the prognostic impact of GDF-15 should be included [Eur Heart J. 2022. doi: 10.1093/eurheartj/ehac577].

6.       The authors conclude that “a higher concentration of GDF-15 on admission can be used as an additional marker of ischemia/reoxygenation injury and subsequent no-reflow phenomenon in patients with STEMI treated with primary PCI”. However, this concept may be true in case the level of GDF-15 can be measured in a timely manner (e.g. within 30 minutes).

Author Response

The authors conducted a prospective, single-center, observational study (n=80), in which the clinical usefulness of GDF-15 was tested. They showed that the higher GDF-15 level was associated with no-reflow phenomenon during the primary PCI procedures. GDF-15 may be a promising biomarker in the field of cardiovascular disease and thus, the present paper may be of clinical interest. However, there are lots of points that should appropriately addressed by the authors. 

Reply: We would like to thank the Reviewer for the time and effort spent on reviewing our manuscript. We have considered these comments very carefully and addressed all of them down below.

Comment 1.
       The authors state that their study is prospectively performed. Thus, the trial registration number should be shown. If not registered, the present study should be considered as if it was a retrospective study. In addition, this reviewer does not understand why the study period was between 2012 and 2014 (i.e. outdated). GDF-15 may have been evaluated using blood samples for some reasons? It is also peculiar that the mean follow-up duration was “only” 461 days against the fact that the study period ended in 2014 (8 years ago). Reply 1: Thank you for this important question, we are glad to explain this issue. Our study was prospective, we recruited 80 consecutive STEMI patients who underwent primary PCI in our centre between 2012 and 2014. GDF-15 concentration was measured on the admission to the hospital in the collected blood samples. We gathered all of the GDF-15 concentrations data in our department’s database. Although we cannot provide the trial registration number as the study was not registered on clinicaltrials.gov at the time, please see that it was approved by the Bioethics Committee of the Medical University of BiaÅ‚ystok (approval no. R-I-002/60/2008 obtained on 28.02.2008, R-I-002/60A/2013 obtained on 19.12.2013). The data on the mortality was obtained over one year after the data gathering. However, due to the grant policy in our institution, we were not able to share our results. Now, we are glad to submit the manuscript to be considered for the publication.

Comment 2.       No-reflow phenomenon in the present study was defined as a corrected TIMI Frame Count excessing 27 and Myocardial Blush Grade less than 3 “after” PCI, and the incidence was reported to be 24%. This reviewer wonders if the primary event included no-reflow phenomenon “during” the primary PCI procedures, or that “after” PCI (i.e. final TIMI flow). If authors included only no-reflow phenomenon “after” PCI, the rate (24%) sounds extremely high. The definition of no-reflow phenomenon, including the timing for evaluation, should be clear.

Reply 2: We thank the Reviewer for the comment. As we agree that 24% of no-reflow phenomenon sounds high, it is consistent with the previous studies. Hibi et al. in their RCT that investigated the utility of distal protection during percutaneous coronary intervention in patients with acute coronary syndromes reported no-reflow phenomenon in 26.5% in the distal protection group and 41.7% in the conventional treatment group (https://doi.org/10.1016/j.jcin.2018.03.021). Rezkalla et al. in their analaysis of 347 patients with ST-segment elevation myocardial infarction treated with primary PCI found that the frequency of no-reflow vary with the method of assessment, ranging from 32% by TIMI to 57% by MBG (10.1111/j.1540-8183.2010.00561.x).

Although the treatment of STEMI is currently well-established, the high prevalence of no-reflow phenomenon despite percutaneous approach suggests that further research should focus on the effective prevention of this dangerous complication.

Comment 3.       The elevated TnI, CK, and CK-MB levels at admission in the no-reflow group may be attributable to the longer onset-to-arrival and onset-to-balloon times. According to the CK level at admission, most of patients in the no-reflow group probably came to the hospital >12 hours after symptom onset. Onset time seems crucial data. Table S2 should be included in the main manuscript. By the way, this reviewer wonders if TnI, CK, and CK-MB levels are expressed as median (IQR), because of the skewed distribution. 

Reply 3: Thank you for this comment. We strongly agree with the Reviewer that no-reflow phenomenon is associated with time. Previous studies reported that prolonged ischemia time is associated with microcirculatory damage and no-reflow phenomenon (Rezkalla et al., https://doi.org/10.1016/j.jcin.2016.11.059; Namazi et al., https://doi.org/10.23750/abm.v92i5.10053). Our results are consistent with these papers. However, our study was focused on the assessment of GDF-15 as a potential predictor of no-reflow phenomenon. In our paper, there was a significant negative correlation between TIMI flow after PCI and GDF-15 concentration and ROC analysis revealed GDF-15 as a predictor of no-reflow. Further studies should evaluate the relation between the time of the symptom onset and GDF-15 concentrations.

As the Reviewer suggested, we included Table S2 in the main manuscript. Moreover, the Discussion section was modified accordingly.

Thank you for the accurate comment regarding results presentation. TnI, CK, and CK-MB levels were expressed as average (SD), now they are presented as median (IQR).

Comment 4.       NS for p values should be avoided. 

Reply 4: Thank you for this comment. Please see the exact p values in the modified Results section.

Comment 5.       A recent, large-scale, patient-level meta-analysis investigating the prognostic impact of GDF-15 should be included [Eur Heart J. 2022. doi: 10.1093/eurheartj/ehac577]. 

Reply 5: We are grateful for this suggestion. We included the meta-analysis in the Discussion section.

Comment 6.       The authors conclude that “a higher concentration of GDF-15 on admission can be used as an additional marker of ischemia/reoxygenation injury and subsequent no-reflow phenomenon in patients with STEMI treated with primary PCI”. However, this concept may be true in case the level of GDF-15 can be measured in a timely manner (e.g. within 30 minutes).

Reply 6: Thank you for this comment. We agree that currently the clinical usefulness of GDF-15 is low. However, there are already rapid bedside tests for the quantitative determination of proteins (e.g., C-reactive protein). Further research may contribute to the development of such tests for GDF-15, as opposed to a time-consuming standard laboratory assay. In the future, this would improve decision-making process and influence rapid initiation of the treatment that protects against no-reflow phenomenon, such as GP IIb/IIIa inhibitors.

Reviewer 2 Report

Thank you for your effort in creating the paper entitled "Growth differentiation factor 15 as a novel predictor of the no-reflow phenomenon in patients with ST-segment elevation myocardial infarction". I have a few remarks: 

  1. To the best of my knowledge GDF 15 factor was previously described in MI patients more than ten years ago, so calling it a novel predictor is an exaggeration. 
  2. The relationship between GDF 15 factor level and AMI outcome was previously described by Wollert et al. (DOI10.1161/CIRCULATIONAHA.106.650846), Khan et al. (DOI 10.1093/eurheartj/ehn600), in STEMI patients treated by PCI - by Dogdu (DOI 10.3390/diseases8020016). The last study is very similar to yours. 
  3. I believe TIMI and MBG score are 4-point classifications (0, 1,2, or 3) and should be presented as categorical variables rather than continuous ones. The same problem is for the NYHA scale in Table S3. Calculating the average for those variables is a major mistake. 
  4. Per the definition, Reflow group TIMI flow should be 3 in all cases, so why is your average 2.98? 
  5. Comparison of the TIMI flow between reflow and no-reflow groups is pointless as you divided patients depending on TIMI flow. 
  6. The causes of long-term mortality are not described; at least, the differentiation between cardiovascular and non-cardiovascular causes would be substantial. 

Author Response

Thank you for your effort in creating the paper entitled "Growth differentiation factor 15 as a novel predictor of the no-reflow phenomenon in patients with ST-segment elevation myocardial infarction". I have a few remarks: 

Reply: We are very grateful for the time and effort invested in reviewing our manuscript. We have considered all comments very carefully and edited the manuscript accordingly to the Reviewer’s suggestions.

Comment 1: To the best of my knowledge GDF 15 factor was previously described in MI patients more than ten years ago, so calling it a novel predictor is an exaggeration. 

Reply 1: Thank you for this observation. The title has been modified accordingly.

Comment 2: The relationship between GDF 15 factor level and AMI outcome was previously described by Wollert et al. (DOI10.1161/CIRCULATIONAHA.106.650846), Khan et al. (DOI 10.1093/eurheartj/ehn600), in STEMI patients treated by PCI - by Dogdu (DOI 10.3390/diseases8020016). The last study is very similar to yours. 

Reply 2: We thank the Reviewer for this comment. We are glad that recent years brought more insight into no-reflow phenomenon pathophysiology. Knowing its prognostic value and poor outcomes despite the guideline-based treatment of ST-elevation myocardial infarction, more impact should be put on its understanding, prevention, and effective treatment. We included the suggested papers in the modified Discussion section.

Comment 3: I believe TIMI and MBG score are 4-point classifications (0, 1,2, or 3) and should be presented as categorical variables rather than continuous ones. The same problem is for the NYHA scale in Table S3. Calculating the average for those variables is a major mistake.

Reply 3: Thank you for the comment. We agree with the Reviewer and apologize for the mistake.

Comment 4: Per the definition, Reflow group TIMI flow should be 3 in all cases, so why is your average 2.98? 

Reply 4: Thank you for this observation. We are glad to explain this issue. In order to use the most objective method, in our study no-reflow phenomenon was defined as the corrected TIMI Frame Count > 27 and Myocardial Blush Grade < 3 after PCI. As Gibson et al. noticed in the original TIMI Frame Count report, although TIMI flow grade is valuable, it is limited by its subjective and categorical nature (https://doi.org/10.1161/01.CIR.93.5.879). Due to the subjective nature of TIMI flow grade, few of our patients not meeting no-reflow definition, were assessed as TIMI flow grade 2 by the physician performing angioplasty and it was reflected in the average of 2.98. In order not to modify the primary assessment we included such patients in our analysis.

Comment 5: Comparison of the TIMI flow between reflow and no-reflow groups is pointless as you divided patients depending on TIMI flow.

Reply 5: We thank the Reviewer for this comment. We agree that our assessment of no-reflow phenomenon was based on the corrected TIMI frame count and Myocardial Blush Grade, therefore there was no use of the comparison between two groups. In accordance to the Reviewer suggestion, we decided to exclude these information from the Table 2.

Comment 6: The causes of long-term mortality are not described; at least, the differentiation between cardiovascular and non-cardiovascular causes would be substantial. 

Reply 6: Thank you for this important comment. We strongly agree that the differentiation between cardiovascular and non-cardiovascular causes of death would be substantial. However, the data on mortality obtained from the Voivodeship Office does not provide information on the cause of death. It was commented in the Limitations section in the previous Polish studies that used similar methodology (Kowalewski et al., https://doi.org/10.3390/jcm9051345). Cause-specific mortality data is available only as an anonymized data set obtained from National Health Fund or National Statistical Office. However, even in such setting the rate of garbage codes might influence our relatively small study group (Kuzma et al., https://doi.org/10.3390/jcm9113445). Please see the updated Limitations section.

Round 2

Reviewer 1 Report

This reviewer feels that the manuscript has improved significantly. In terms of the former comment #2, however, no-reflow phenomenon "during" or "after" PCI should be clear.

Author Response

This reviewer feels that the manuscript has improved significantly. In terms of the former comment #2, however, no-reflow phenomenon "during" or "after" PCI should be clear. 

Reply: We are grateful for the appreciation of our changes and the time invested in the review.

The aim of our study was to evaluate the prognostic value of GDF-15 concentration on the disturbances affecting microcirculatory reflow after primary percutaneous coronary intervention. Therefore, we defined no-reflow phenomenon as a corrected TIMI Frame Count excessing 27 and Myocardial Blush Grade less than 3 immediately after angioplasty. We have modified the Material and methods section accordingly. We hope that our changes make this issue more clear.

Reviewer 2 Report

Thank you for considering my suggestions. I have reread the manuscript very carefully. 

Here are a couple of remarks: 

  1. I believe the multivariate logistic regression analysis is presented in Table 4 (please change the number in line 209 and the table's title accordingly). 
  2. I'm afraid I have to disagree with your arguments concerning the causes of not collecting specific data about mortality causes. Firstly, you enrolled only 80 patients and described your study as 'prospective'. In contrast, Kowalewski et al. checked 193,488 patients undergoing CABG surgery in KROK, then used PS and finally analyzed 864 pts in his observational study, so ten times more than you. When conducting the prospective study on a small group of patients, you can efficiently perform telephone follow-ups and contact families about the causes of death (only 8% of your patients, so around six people). Secondly, your mean follow-up was 461 days, so about 1.5 years - not so long to be unable to contact the patients. Finally, you have 3% of in-hospital mortality - I assume you can access the documentation and assess the cause of death in those individuals. 
  3. I checked the mortality percentages you provided and need help understanding line 195: you wrote that total mortality was 3% for in-hospital (out of 80 patients it is 2.4) and 8% for long-term (out of 80 it is 6.4). You should be more precise here. 
  4. I am sorry, but I do not understand the argument about garbage codes and the connection between your study and the one of Kuzma et al. 

Author Response

Thank you for considering my suggestions. I have reread the manuscript very carefully. 

Reply: We are once again grateful for the time and effort invested in the review. We have considered the comments very carefully and addressed all of them down below.

Here are a couple of remarks: 

Comment 1: I believe the multivariate logistic regression analysis is presented in Table 4 (please change the number in line 209 and the table's title accordingly). 

Reply 1: Thank you for the remark. We apologize for the mistakes. We modified our manuscript accordingly.

Comment 2: I'm afraid I have to disagree with your arguments concerning the causes of not collecting specific data about mortality causes. Firstly, you enrolled only 80 patients and described your study as 'prospective'. In contrast, Kowalewski et al. checked 193,488 patients undergoing CABG surgery in KROK, then used PS and finally analyzed 864 pts in his observational study, so ten times more than you. When conducting the prospective study on a small group of patients, you can efficiently perform telephone follow-ups and contact families about the causes of death (only 8% of your patients, so around six people). Secondly, your mean follow-up was 461 days, so about 1.5 years - not so long to be unable to contact the patients. Finally, you have 3% of in-hospital mortality - I assume you can access the documentation and assess the cause of death in those individuals. 

Reply 2: Thank you for the comment. We agree with the Reviewer that our study included the small number of the individuals that could have been potentially followed-up for the cause of death, however, there are some issues that may not have been clarified before. The main aim of our study was to assess the correlation between GDF-15 concentration and disturbances affecting microcirculatory reflow after primary percutaneous coronary intervention. The ROC analysis confirmed GDF-15 as a predictor of no-reflow phenomenon. Relation between GDF-15 and mortality was the secondary endpoint of our study and due to the study design, we did not perform either the clinical or telephone follow-up. According to the Polish Statistical Office, in 2015 garbage codes were used in over 30% of all deaths (Statistics Poland, data available only in Polish https://stat.gov.pl/obszary-tematyczne/ludnosc/statystyka-przyczyn-zgonow/zgony-wedlug-przyczyn-okreslanych-jako-garbage-codes,3,3.html). Moreover, the number of autopsies in Poland is steadily decreasing due to the financial issues. Therefore, even if the family of the patient had been reached, the claimed cause of death may not be the actual cause of death.
When it comes to the in-hospital deaths, all of them were cardiovascular deaths.

Comment 3: I checked the mortality percentages you provided and need help understanding line 195: you wrote that total mortality was 3% for in-hospital (out of 80 patients it is 2.4) and 8% for long-term (out of 80 it is 6.4). You should be more precise here. 

Reply 3: Thank you for the suggestion. The mortality rates were rounded to the nearest integer. According to the Reviewer’s suggestion, exact numbers are now provided in the Results section.

Comment 4: I am sorry, but I do not understand the argument about garbage codes and the connection between your study and the one of Kuzma et al. 

Reply 4: Thank you for the comment. We apologize for the lack of precision. We cited two papers conducted in Poland to emphasize that the cause-specific mortality in our country is an issue that have been faced by other researchers as well. As we have replied to the comment #2, in 2015 garbage codes were used in over 30% of all deaths and due to the fact that National Health Fund does not cover the costs of autopsies, their number is steadily decreasing (https://stat.gov.pl/obszary-tematyczne/ludnosc/statystyka-przyczyn-zgonow/zgony-wedlug-przyczyn-okreslanych-jako-garbage-codes,3,3.html). In the last years various systemic changes have been implemented and the number of garbage codes has decreased from 32.3% in 2015 to 28.5% in 2019. We are looking forward to reach the rates of garbage codes reported in the UK or Sweden (22.09% and 24.85% in 2015, respectively; Monasta et al. https://doi.org/10.1093/eurpub/ckac021). We hope that the issue of high rate of garbage codes in Poland is more clear now.